# Higher-Order Optomechanical Nonlinearity Based on the Mechanical Effect of Light

**Qin Wu [1,\*] and Hao-Jin Sun [2]**

1    School of Biomedical Engineering, Guangdong Medical University, Dongguan 523808, China
2    School of Electronic Engineering & Intelligentization, Dongguan University of Technology,
      Dongguan 523808, China; haojin_sun@dgut.edu.cn
\*    Correspondence: wuqin@gdmu.edu.cn

**Abstract:** Nonlinear cavity optomechanics based on the mechanical effect of light has recently received considerable attention due to its potential applications in high-precision metrology. In this work, we theoretically studied the third-order optomechanical nonlinearity by using a perturbative approach, and an analytical solution is given, which can be extended to cases of higher-order optomechanical nonlinearity. Furthermore, the generation of a third-order sideband is analyzed in detail, and the results show that the amplitude of the third-order sideband shows a high dependence on the control field detuning, suggesting that the high-order nonlinear intensity can be enhanced by properly adjusting the detuning of the laser field rather than by a strong laser drive. In addition to providing insight into optomechanical nonlinearity, the analytical description of third-order optomechanical nonlinearity based on the mechanical effects of light may find applications in ultra-high precision measurement under low power conditions.

**Keywords:** cavity optomechanics; optomechanical nonlinearity; third-order sideband; perturbative approach

## 1. Introduction

Cavity optomechanics studying the interaction between light and mechanical oscillation has recently become a rapidly developing field and now plays an important role in many fields of physics, including cooling of mechanical oscillators [1–3], gravitational-wave detectors [4], manipulation of light propagation [5–7], integrated optical components [8], and so on [9]. Numerous studies have shown that many interesting effects uncovered in atomic-molecular systems can also be observed in the optomechanical system through mechanical effects of light. A classical example is optomechanically induced transparency/absorption (OMIT), which has been predicted theoretically [10–12] and verified experimentally [13–15]. OMIT, a direct analog of electromagnetically induced transparency, is a kind of induced transparency caused by the radiation pressure of coupling light to mechanical oscillator modes, in which the transmission of a probe field can be regulated all-optically by using a strong driving field, and can be well described through the linearization of optomechanical interactions. More specifically, the optomechanical system is pumped by a strong driving field with frequency $\omega_l$ and a weak detection field with frequency $\omega_p$. Then, a spectrum of frequency $\omega_l \pm n\Omega$ appears in the output field, where $n$ is an integer representing the $n$-order sideband [16–20]. For example, the output fields with frequencies $\omega_l \pm 1\Omega$ and $\omega_l \pm 2\Omega$ denote the first- and second-order upper or lower sideband [16], respectively. In particular, based on the linearized dynamics of the optomechanical interactions $\omega_l \pm 1\Omega$, a transparent window for the propagation of the probe field is induced by the incident field when the resonance condition is satisfied. It has been shown that this intriguing phenomenon provides a unique platform for achieving precision measurement [21–29], such as precision measurement of electrical charges with OMIT [23] and mass sensor with an optomechanical microresonator [24].

Nonlinear cavity optomechanics has recently been the topic of widespread investigations and has developed enormously over the past decades. Many interesting phenomena deriving from the nonlinear optomechanical interactions have been uncovered [30–37], ranging from higher-order sidebands generation [19,20,26] and cavity optomechanical chaos [33,36] to the Kuznetsov–Ma soliton in a microfabricated optomechanical array [32]. An analytical method of describing the nonlinear optomechanical interaction has been proposed. One of the most typical examples is the second-order sideband generation [16] when the nonlinear term of the system was taken into account. Furthermore, the traditional optomechanical system is further extended to the field of spintronics. Specifically, the generalized optomechanical system, which has a similar form of radiation-pressure-type interaction, and the interesting physical phenomenon of the sideband comb as well as the analogous laser action of magnons have been observed in the field of magnonics [38–40]. Moreover, a large number of studies have shown that nonlinear interaction between cavity fields and mechanical oscillation in the optomechanical system may provide metrology with a higher precision and require less power [28]; for example, the precision measurement of electrical charges beyond linearized dynamics [25,27], which can achieve the accuracy of measuring a single charge.

With the requirement of ultra-weak power and more accurate measurement, it is necessary to study the high-order nonlinear effects in the cavity optomechanical system. In this work, the third-order optomechanical nonlinearity based on the mechanical effect of light has been theoretically analyzed, which goes beyond the conventional linearized description of optomechanical interactions. Furthermore, the generation of a third-order sideband is analyzed in detail; that is, the sideband generation process with frequency component $\omega_l \pm 3\Omega$ from the output field, while $\omega_l + 3\Omega$ is the third-order upper sideband and $\omega_l - 3\Omega$ is the lower sideband. Physically, the generation of the high-order sideband is essentially the production and absorption process of multiple phonons caused by the nonlinear term of optomechanical interaction, which is consistent with the physical process of higher harmonic generation caused by the nonlinear term of interaction between light and atoms in atomic–molecular systems (i.e., the production and absorption process of multiple photons) [41]. Therefore, the study of the high-order sideband is of great significance for understanding the nonlinear characteristics of optomechanical interaction and its related applications. Here, we only focus on third-order upper sideband generation and give its analytic expression by way of the perturbation method. We find that the amplitude of the third-order sideband can be substantively modified by OMIT where the spectrum of the third-order sideband is exactly the opposite of the OMIT spectrum. In addition, the relationship between the amplitude of the third-order sideband and the control field detuning $\Delta_c$ under the different driven frequency $\Omega$ was discussed in detail. We believe that the research of the third-order sideband generation will provide good theoretical guidance for the fundamental investigation of nonlinear cavity optomechanics [42] and offer a nonlinear optical method with more accurate precision measurements.

The rest of this paper is organized as follows. In Section 2, we introduce the physical patter of a traditional cavity optomechanical system and give the derivation of the Heisenberg–Langevin equation of motion in the presence of a strong pumping field and a weak probe field. Moreover, the third-order sideband generation induced by the higher-order optomechanical nonlinearity is discussed and the analytical expression is given. In Section 3, the variation of the third-order sideband generation efficiency with the power of the control field is discussed in detail. Furthermore, we show that control field detuning plays an important role in the generation of the third-order sideband. Finally, a conclusion is presented in Section 4.

## 2. Model and Dynamics

The physical pattern we consider is a traditional cavity optomechanical system, as diagrammatically shown in Figure 1. The system consists of a hight-Q Fabry-Pérot cavity, in which one mirror is fixed and the other is movable and treated as a mechanical

oscillator with effective mass $m$ and vibration frequency $\Omega_m$. The single-mode cavity field with eigenfrequency $\omega_c$ couples to the mechanical oscillator via optical radiation pressure. The Hamiltonian of this cavity optomechanical system, without loss of generality, reads as follows [9]:

$$\hat{H}_0 = \frac{\hat{p}^2}{2m} + \frac{1}{2}m\Omega_m^2\hat{x}^2 + \hbar\omega_c\hat{a}^\dagger\hat{a} + \hbar G\hat{x}\hat{a}^\dagger\hat{a}, \tag{1}$$

where the first two terms give the free Hamiltonian of the mechanical oscillator with $\hat{x}$ and $\hat{p}$ being the position and momentum operators of the movable mirror, respectively. The third term denotes the Hamiltonian of the cavity, in which the operators $\hat{a}$ and $\hat{a}^\dagger$ are the bosonic annihilation and creation operators, which obey the commutation relations $[\hat{a}_j, \hat{a}_{j'}^\dagger] = \delta_{jj'}$, $[\hat{a}_j, \hat{a}_{j'}] = [\hat{a}_j^\dagger, \hat{a}_{j'}^\dagger] = 0$. The last term describes the Hamiltonian interaction between the cavity field and the movable mirror with coupling strength G. Here, we presume that the optomechanical system is driven by a strong pumping field with frequency $w_l$ and a weak probe field with frequency $w_p$. Therefore, the Hamiltonian of the driven optomechanical system can be written as follows:

$$\begin{aligned}\hat{H} = \quad &\hat{H}_0 + i\hbar\sqrt{\eta_c\kappa}(\varepsilon_l e^{-i\omega_l t}\hat{a}^\dagger - \varepsilon_l^* e^{i\omega_l t}\hat{a}) \\ &+ i\hbar\sqrt{\eta_c\kappa}(\varepsilon_p e^{-i\omega_p t}\hat{a}^\dagger - \varepsilon_p^* e^{i\omega_p t}\hat{a}).\end{aligned} \tag{2}$$

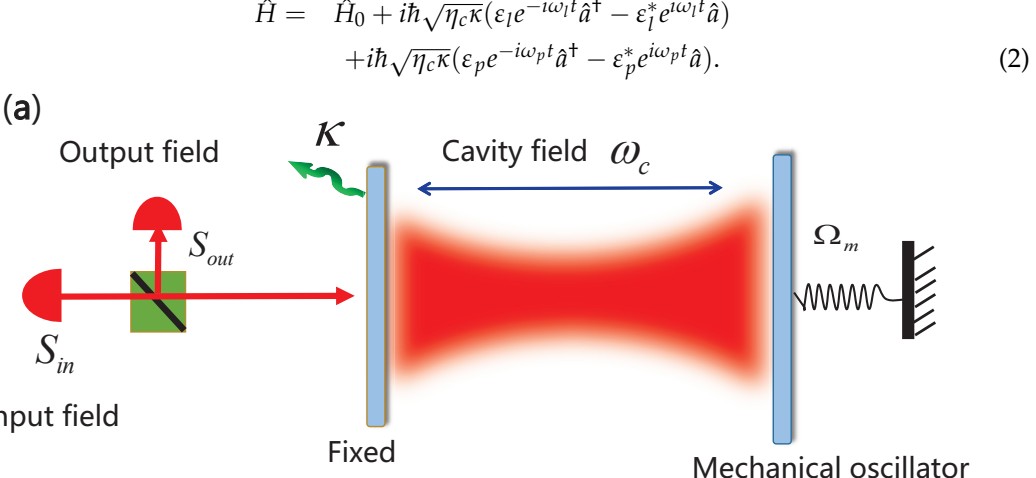

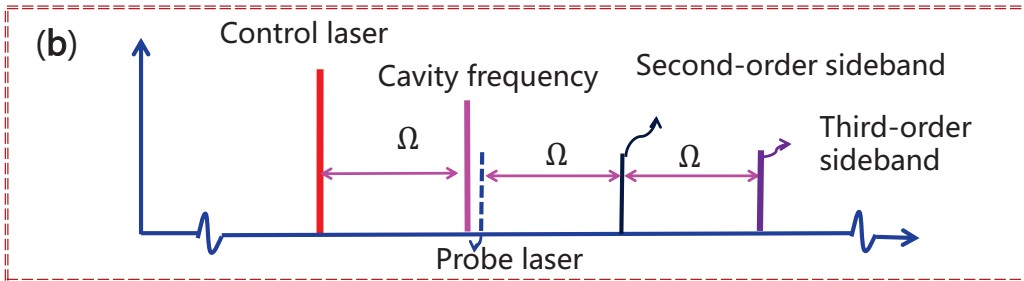

**Figure 1.** (**a**) The physical diagram of a traditional optomechanical system, in which one mirror is fixed and the other is movable and treated as a mechanical oscillator. The system is pumped by a strong input field $S_{in}$ and the output field $S_{out}$ of the system can be analyzed by a spectrum detector. $\kappa$, $\omega_c$, and $\Omega_m$ represent the dissipation of the cavity field, the eigenfrequency of the cavity field, and the intrinsic frequency of the mechanical oscillator, respectively. (**b**) Frequency spectrum of the optomechanical system. The red line represents the control laser, which is detuned by $\Omega$ from the cavity frequency. Second-, third- and high-order sidebands appear in the transmission spectrum.

Here, $\eta_c$ represents the coupling parameter between the external input field and the cavity field, whose coupling value is selected as 1/2, and are used throughout the whole work. $\varepsilon_i = \sqrt{P_i/\hbar\omega_i}(i = l, p)$ are the amplitudes of the input field with $P_l$ representing the pump power of the control field, $P_p$ the power of the probe field and $\kappa$ the total decay rate of the cavity. In the rotating reference frame with frequency $w_l$ of the input laser based on a unitary transformation $U(t) = \exp[-iw_c\hat{a}^\dagger\hat{a}t]$, the system Hamiltonian in Equation (2) becomes

$$\hat{H} = \frac{\hat{p}^2}{2m} + \frac{1}{2}m\Omega_m^2\hat{x}^2 + \hbar\Delta_c\hat{a}^\dagger\hat{a} + \hbar G\hat{x}\hat{a}^\dagger\hat{a}$$
$$+i\hbar\sqrt{\eta_c\kappa}[(\varepsilon_l + \varepsilon_p e^{-i\Omega t})\hat{a}^\dagger - H.c.]. \tag{3}$$

where $\Delta_c = \omega_l - \omega_c$ is the detuning between the cavity field frequency and the driving field frequency, and $\Omega = \omega_p - \omega_l$ is the beat frequency between the driving field and the probe field. After introducing the dissipation and fluctuation terms with the Markov approximation [13], the dynamic evolution of the cavity optomechanical system governed by the Hamiltonian in Equation (3) can be described by the following Heisenberg–Langevin equations:

$$\diamond \cdot \mho = M\mho + \eth + \Gamma, \tag{4}$$

where $\diamond \cdot \mho = (d\hat{a}/dt, \hat{\Psi}\hat{x})^T$ with $\hat{\Psi} = m(d^2/dt^2 + \gamma_m d/dt + \Omega_m^2)$, $\mho = (\hat{a}, \hat{x})^T$, $\eth = (\sqrt{\eta_c\kappa}(\varepsilon_l + \varepsilon_p e^{-i\Omega t}), 0)$, $\Gamma = (\hat{a}_{in}(t), \hat{\xi}(t))^T$ and

$$M = \begin{pmatrix} i\Delta_c - iG\hat{x} - \kappa/2 & 0 \\ -\hbar G\hat{a}^\dagger & 0 \end{pmatrix}.$$

Here, $\gamma_m$ indicates the dissipation rating of the mechanical oscillator, which is introduced phenomenologically. The noise matrix $\Gamma = [\hat{a}_{in}(t), \hat{\xi}(t)]^T$ with operators $\hat{a}_{in}(t)$ and $\hat{\xi}(t)$ denote environmental noises corresponding to the operators $\hat{a}$ and $\hat{p}$. Here, we can assume that the quantum noise $\hat{a}_{in}(t)$ has zero mean values, based on the nonvanishing commutation relations $\langle a_{in}(t)a_{in}^\dagger(t')\rangle = \delta(t - t')$ and $\langle a_{in}^\dagger(t)a_{in}(t')\rangle = 0$. Furthermore, the thermal Langevin force $\hat{\xi}(t)$ also has a vanishing mean value $\langle\xi(t)\rangle = 0$, resulting from the temperature-dependent correlation function $\langle\xi(t)\xi^\dagger(t')\rangle = \gamma_m(2\pi w_m)^{-1}\int\exp[-iw(t - t')][1 + \coth(\hbar w/2k_B T)]dw$ [43], where $k_B$ and T are the Boltzmann constant and the temperature of the reservoir of the mechanical oscillator, respectively.

Equation (4) can be solved by using the perturbation method. By using $a = \bar{a} + \delta a$ and $x = \bar{x} + \delta x$ where

$$\bar{a} = \frac{\sqrt{\eta_c\kappa}\varepsilon_l}{(-i\bar{\Delta} + \kappa/2)}, \qquad \bar{x} = \frac{-\hbar G|\bar{a}|^2}{m\Omega_m^2}, \tag{5}$$

with $\bar{\Delta} = \Delta - G\bar{x}$, are the static solutions of the cavity field and the mechanical oscillator displacement for the case where the driving field is much stronger than the probe field and where all time derivatives vanish. Correspondingly, $\delta a$ and $\delta x$ are the fluctuations around the steady-state solutions of the cavity field $a$ and the mechanical displacement $x$, respectively. After a simple calculus, we can obtain the nonlinear matrix equation that $\delta a$ and $\delta x$ satisfy, as shown above (under the mean-field approximation)

$$\diamond \cdot \Phi = \mu\Phi + \nu\Phi^* + \sqrt{\eta_c\kappa}\varepsilon_p e^{-i\Omega t}\%, \tag{6}$$

where $\diamond \cdot \Phi = (d\delta a/dt, \hat{\Psi}\delta x)^T$, $\Phi = (\delta a, \delta x)^T$, $\% = (1,0)^T$, and

$$\mu = \begin{pmatrix} i\bar{\Delta} - \kappa/2 & -iG(\bar{a} + \delta a) \\ \hbar G(\bar{a}^* + \delta a^*) & 0 \end{pmatrix}, \quad \nu = \hbar G\begin{pmatrix} 0 & 0 \\ \bar{a} & 0 \end{pmatrix}.$$

Making the following ansatz [44],

$$\delta a = A_1^- e^{-i\Omega t} + A_1^+ e^{i\Omega t} + A_2^- e^{-i\Omega t} + A_2^+ e^{i\Omega t} + A_3^- e^{-i\Omega t} + A_3^+ e^{i\Omega t},$$
$$\delta x = X_1 e^{-i\Omega t} + X_1^* e^{i\Omega t} + X_2 e^{-i\Omega t} + X_2^* e^{i\Omega t} + X_3 e^{-i\Omega t} + X_3^* e^{i\Omega t}. \tag{7}$$

Substituting such ansatz into Equation (6), we can get three sets of nonlinear matrix equations regarding the amplitude of the sidebands:

$$\Omega_n \cdot A_n = \rho \cdot \sigma_n + U, \tag{8}$$

where $A_n = (A_n^-, A_n^+, X_n)^T$ and $n = 1, 2, 3$ describe the first-, second-, and third-order sideband, respectively. Here, we should note that $U = (-\sqrt{\eta_c \kappa} \varepsilon_p, 0, 0)$ when $n = 1$, in other cases, $U = (0, 0, 0)$. The coefficient matrix

$$\Omega_n = \begin{pmatrix} \Theta + ni\Omega & 0 & 0 \\ 0 & \Theta - ni\Omega & 0 \\ 0 & 0 & \Re_n \end{pmatrix}, \quad \rho = \begin{pmatrix} iG & 0 & 0 \\ 0 & iG & 0 \\ 0 & 0 & -\frac{\hbar G}{m} \end{pmatrix}.$$

with $\Theta = i\Delta_c - iG\bar{x} - \kappa/2$ and $\Re_n = \Omega_m^2 - (n\Omega)^2 - ni\Gamma_m\Omega$. The matrix $\sigma_n$ directly determines the magnitude of the effective sidebands amplitude and we call it the sideband matrix element.

$$\sigma_1 = \begin{pmatrix} \bar{a}X_1 \\ \bar{a}X_1^* \\ \bar{a}(A_1^+)^* + \bar{a}^* A_1^- \end{pmatrix}, \quad \sigma_2 = \begin{pmatrix} \bar{a}X_2 + X_1 A_1^- \\ \bar{a}X_2^* + X_1^* A_1^+ \\ \bar{a}(A_2^+)^* + \bar{a}^* A_2^- + (A_1^+)^* A_1^- \end{pmatrix}$$

describe the effect of OMIT and the second-order sideband, respectively, which have been obtained in previous work [16], and

$$\sigma_3 = \begin{pmatrix} \bar{a}X_3 + X_1 A_2^- + X_2 A_1^- \\ \bar{a}X_3^* + X_1^* A_2^+ + X_2^* A_1^+ \\ \bar{a}(A_3^+)^* + \bar{a}^* A_3^- + (A_1^+)^* A_2^- + (A_2^+)^* A_1^- \end{pmatrix}$$

describes the effect of the third-order sideband of such an optomechanical system, which has not yet been studied.

Equations (8) can easily be solved and $A_1^-$, $A_2^-$ and $A_3^-$ are obtained as follows:

$$A_1^- = \frac{-B^*(\Omega)\sqrt{\eta_c \kappa} \varepsilon_p}{E(\Omega)},$$

$$A_2^- = \frac{D(2\Omega)(A_1^+)^* A_1^- - C(2\Omega)X_1(A_1^+)^* + iGB^*(2\Omega)A_1^- X_1}{E(2\Omega)},$$

and

$$A_3^- = \frac{D(3\Omega)[(A_2^+)^* A_1^- + A_2^-(A_1^+)^*] - C(3\Omega)[X_1(A_2^+)^* + X_2(A_1^+)^*] + iGB^*(3\Omega)\aleph}{E(3\Omega)},$$

where $\aleph = X_1 A_2^- + X_2 A_1^-$ and

$$A_1^+ = \frac{iG\bar{a}X_1^*}{\Theta - i\Omega}, \quad X_1 = \frac{(\Theta + i\Omega)A_1^- + \sqrt{\eta_c \kappa} \varepsilon_p}{iG\bar{a}},$$

$$A_2^+ = \frac{iG(\bar{a}X_2^* + X_1^* A_1^+)}{\Theta - 2i\Omega}, \quad X_2 = \frac{(\Theta + 2i\Omega)A_2^- - iGX_1 A_1^-}{iG\bar{a}},$$

and

$$\chi(\Omega) = 1/m(\Omega_m^2 - \Omega^2 - i\Gamma_m\Omega),$$
$$B(\Omega) = \Theta - i\Omega + i\hbar G^2\chi(-\Omega)|\bar{a}|^2,$$
$$C(\Omega) = \hbar G^3\chi(\Omega)\bar{a}^2,$$
$$D(\Omega) = \hbar G^2\chi(\Omega)\bar{a}(\hbar G^2\chi(\Omega)|\bar{a}|^2 - iB^*(\Omega)),$$
$$E(\Omega) = B(-\Omega)B^*(\Omega) - \hbar^2 G^4\chi^2(\Omega)|\bar{a}|^4. \tag{9}$$

From Equation (9), we can clearly get the mechanism of the sidebands generation: the first-order sideband is proportional to the amplitude of the probe field and the second-order sideband is mainly generated from the first-order sideband, and the third-order sideband is induced both by the first- and second-order sidebands. As a result, the effective sideband

strength will be weaker and weaker, so it is particularly important to improve the intensity of the higher-order sideband generation.

The output field from the cavity optomechanical system can be acquired by using the standard input–output relationship [45]:

$$
\begin{aligned}
S_{out} =& S_{in} - \sqrt{\eta_c \kappa} a \\
=& (\varepsilon_l - \sqrt{\eta_c \kappa} \bar{a}) e^{-i\omega_l t} + (\varepsilon_p - \sqrt{\eta_c \kappa} A_1^-) e^{-i(\omega_l + \Omega)t} \\
& - \sqrt{\eta_c \kappa} A_1^+ e^{-i(\omega_l - \Omega)t} - \sqrt{\eta_c \kappa} A_2^- e^{-i(\omega_l + 2\Omega)t} \\
& - \sqrt{\eta_c \kappa} A_2^+ e^{-i(\omega_l - 2\Omega)t} - \sqrt{\eta_c \kappa} A_3^- e^{-i(\omega_l + 3\Omega)t} \\
& - \sqrt{\eta_c \kappa} A_3^+ e^{-i(\omega_l - 3\Omega)t}
\end{aligned}
\tag{10}
$$

The terms $(\varepsilon_p - \sqrt{\eta_c \kappa} A_1^-) e^{-i(\omega_l + \Omega)t}$ and $-\sqrt{\eta_c \kappa} A_1^+ e^{-i(\omega_l - \Omega)t}$ describe the first-order upper sideband and lower sideband, respectively. The terms $-\sqrt{\eta_c \kappa} A_2^- e^{-i(\omega_l + 2\Omega)t}$ and $-\sqrt{\eta_c \kappa} A_2^+ e^{-i(\omega_l - 2\Omega)t}$ express the second-order upper sideband and lower sideband, respectively. Without losing generality, the third-order upper sideband and lower sideband correspond to the term $-\sqrt{\eta_c \kappa} A_3^- e^{-i(\omega_l + 3\Omega)t}$ and $-\sqrt{\eta_c \kappa} A_3^+ e^{-i(\omega_l - 3\Omega)t}$, respectively.

The transmission of the probe field can be defined as the ratio between the amplitude of the output field and the probe field, as follows

$$
\begin{aligned}
t_p =& \frac{(\varepsilon_p - \sqrt{\eta_c \kappa} A_1^-)}{\varepsilon_p} \\
=& 1 + \frac{B^*(\Omega)}{E(\Omega)} \eta_c \kappa,
\end{aligned}
\tag{11}
$$

By the same token, we can define a dimensionless expression

$$
\eta = \left| \frac{-\sqrt{\eta_c \kappa} A_3^-}{\varepsilon_p} \right|,
\tag{12}
$$

as the efficiency of the third-order sideband generation.

## 3. Results and Discussion

In the following section, we turn to discuss how the efficiency of the third-order upper sideband varies with the optical power of the control field $P_l$. After such discussion, we find that the efficiency of the third-order sideband process can be substantively modified by the control field. And more importantly, the amplitude of the third-order sideband can be enhanced by tuning the driven field detuning $\Delta_c$. The simulation parameters used in this work are the effective mass of the mechanical oscillator $m = 20$ ng, the vibration frequency of the mechanical oscillator $\Omega_m/2\pi = 51.8$ MHz, the decay rate of the mechanical oscillator $\gamma_m/2\pi = 41.0$ kHz, the optomechanical coupling strength $G/2\pi = -12$ GHz/nm, the detuning of the cavity field $\Delta_c = -\Omega_m$, the total loss rate of the cavity field $\kappa/2\pi = 15.0$ MHz, and the drive field with wavelength $\lambda_c = 2\pi c/\omega_c = 532$ nm. These parameters are chosen from the experiment parameters [13], and are used throughout the whole work.

The efficiency of the third-order sideband process $\eta$ varies with the driving frequency $\Omega$ and the power of control field $P_l$ is plotted in Figure 2. Here, we should note that the efficiency of the third-order sideband was used as $\eta(100\%)$. As shown in Figure 2, the depth of the color represents the efficiency of third-order sideband process $\eta(100\%)$. The efficiency of the third-order sideband almost tends to zero near the resonance condition $\Omega = \Omega_m$ (middle blue area). There is a transparent window under the resonance condition $\Omega = \Omega_m$ and the probe light is completely transmitted, which leads to no excess light with which to induce the third-order sideband generation, so the efficiency of $\eta$ is extremely weak. On both sides of $\Omega/\Omega_m = 1$, however, the efficiency of the third-order sideband is enhanced, which means that there are two absorption valleys in the transmission spectrum.

Under the circumstances, the probe field is almost completely absorbed and induces the generation of the third-order sideband. From above analysis, we can know that the spectral distribution of the third-order sideband is completely opposite to the first-order sideband.

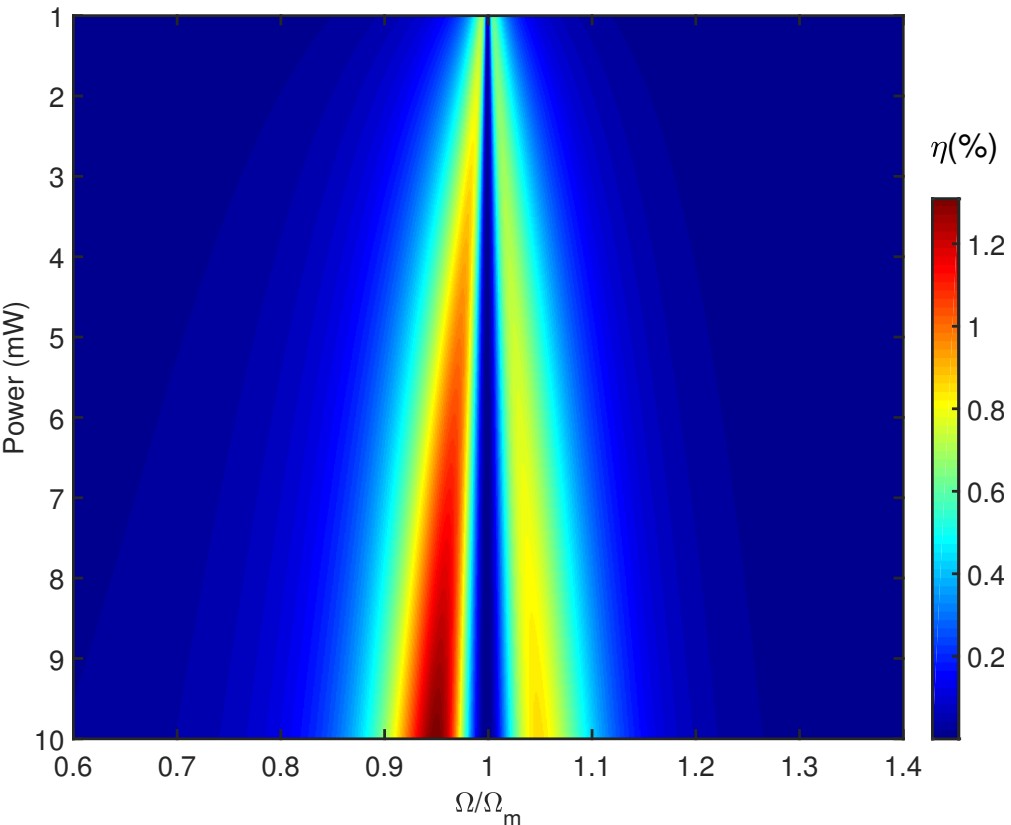

**Figure 2.** The efficiency of the third-order sideband generation $\eta(\%)$ varies with the optical power of the control field $P_l$ and the beat frequency $\Omega/\Omega_m$. The parameters are $m = 20$ ng, $G/2\pi = -12$ GHz/nm, $\Omega_m/2\pi = 51.8$ MHz, $\gamma_m/2\pi = 41.0$ kHz, $\Delta_c = -\Omega_m$, $\kappa/2\pi = 15.0$ MHz, and $\varepsilon_p = 0.05\varepsilon_l$ [13].

Hence, we call it the optomechanically induced opacity of the third-order sideband, which means that the third-order sideband and even the higher-order sidebands are derived from the first-order sideband. From Figure 2, we can see that the efficiency of third-order sideband generation increases as the optical power of the control field increases and the maximum value of $\eta$ is about 1.2%, corresponding to the control field 10 mW. In addition, the width of the opacity window increases as the optical power of the control field increases, which coincides with the regular pattern of the first-order sideband varying with the power of the driven field.

A high dependence of the efficiency of third-order sideband generation on the pumping field power is observed and more exact results are plotted in Figure 3 by using the same experiment parameters [13]. Due to the weakness of power of the control field ($P_l = 0.1$ mW), the efficiency of $\eta$ is very low, about 0.12% under the condition $\Omega/\Omega_m = 1$, as shown in Figure 3a. Moreover, from the screenshot of Figure 3a, we can see that there is not the optomechanically induced opacity of the third-order sideband, which means that most of the probe field is absorbed. As expected, the efficiency of $\eta$ is enhanced when we increase the pumping field power. Figure 3b shows that the efficiency of third-order sideband generation increases to about 0.4% in the case of the control field power $P_l = 0.1$ mW. Understandably, with the increase of the drive field power, the photon number of the intracavity field increases, and the nonlinear response of the system is also enhanced. Meanwhile, there is an opacity window near the resonance condition $\Omega = \Omega_m$, although it is not deep. The appearance of the opaque window can be explained by the splitting of the system energy level. On the one hand, when there is no control field, or the control field is weak,

the coupling between the mechanical oscillator and the cavity field will form two dressed levels, and an obvious absorption peak will appear in the transmission spectrum under the resonance condition. On the other hand, when the power of the control field increases gradually, one of the dressed energy levels will be modified and energy-level splitting will occur, and the absorption peak will gradually disappear under the resonance condition, thus forming a transparent window, namely OMIT [10]. At this time, there is not enough energy to stimulate the generation of the third-order sideband. Consequently, an opaque window will appear in the third-order sideband generation spectrum. To further enhance the efficiency of third-order sideband generation, we increase the power of the pumping field $P_l$ = 1 mW, and the result is shown in Figure 3c) It is clearly seen that the opacity window near the resonance condition $\Omega/\Omega_m = 1$ becomes distinct relative to the case in Figure 3b. In addition, the efficiency of the third-order sideband $\eta$ increases with the power of the driven field increasing and the maximum value of $\eta$ is about 0.7%. Subsequently, we increase the control field power again to $P_l$ = 10 mW, as shown in Figure 3d. Visibly, we can see that not only the maximal efficiency of the third-order sideband increases to about 1.4%, but also the width of the opacity window increases significantly. Considering Figures 2 and 3 together, if the power of the control field $P_l$ is asthenic, OMIT will not appear in the output spectrum. In this case, around the resonance condition, i.e., $\Omega = \Omega_m$, the transmission coefficient of the probe field is almost zero, which means that the probe field is almost completely absorbed. Meanwhile, the third-order sideband $\eta$ achieves the maximum amplitude at the resonance condition $\Omega = \Omega_m$. However, when we increase the pumping field power, the transparent window takes place near the resonance condition $\Omega = \Omega_m$. From the above discussion, we can note that the efficiency of the third-order sideband is more sensitive than OMIT with the changes of the control field, whether it is the range of intensity changes, or the width of the opacity window. Based on this, we can use the effect of the nonlinear third-order sideband for more accurate precision measurement [25,27].

For more intuitive study of the influence of OMIT to the efficiency of the third-order sideband, the calculation results of $|t_p|^2$ and $\eta$ vary with the power of the control field at the resonance condition $\Omega/\Omega_m = 1$ and are plotted in Figure 4. The red solid line represents the change of $|t_p|^2$ varying with the control field and the pink dotted line represents the efficiency of third-order sideband generation. Clearly, we can see that $|t_p|^2$ increases as the power of the control field increases, while the efficiency of third-order sideband $\eta$ decreases rapidly after a rapid as well as brief increase, and finally tends to zero. More specifically, when the control field is weaker than about 0.1 mW, $\eta$ increases sharply with the optical power of the control field and reaches its maximum at about $P_l$ = 0.1 mW. For another case where the optical power of the control field is larger than 0.1 mW, $|t_p|^2$ increases continuously and finally stabilizes while $\eta$ decreases slowly and finally stabilizes quite low. The reason for this phenomenon is that under the action of a strong control field, the appearance of OMIT suppresses the generation of the third-order sideband under the resonance condition $\Omega/\Omega_m = 1$.

Up to now, we have mainly focused on the influence of OMIT on the third-order sideband generation. Now, however, we turn to discuss how to increase the amplitude of the third-order sideband by regulating the detuning between the cavity field and the control field $\Delta_c$. The efficiency of the third-order sideband process $\eta$ varies with the driving frequency $\Omega$ and the detuning $\Delta_c$ is plotted in Figure 5. Here, the pumping field power used is $P_l$ = 5 mW. In the case of $\Delta_c = \Omega_m$, the efficiency of the third-order sideband $\eta$ is only about 1%, the same as in Figure 2. With the change of the control field detuning, the amplitude of the third-order sideband has greatly increased. Specifically, the amplitude of the third-order sideband $\eta$ increases to about 5% when the control field detuning is taken to $\Delta_c \approx -1.2\Omega_m$.

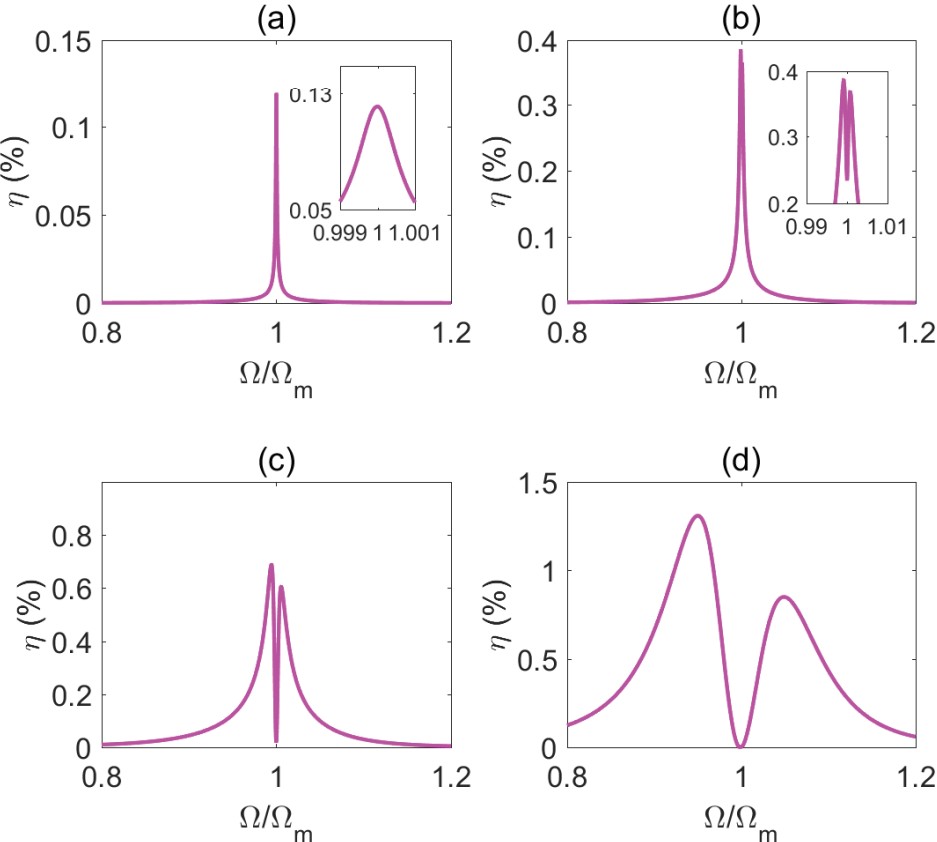

**Figure 3.** The efficiency of the third-order sideband generation $\eta(\%)$ as a function of the beat frequency $\Omega/\Omega_m$ under different control fields. The powers of the control fields are (**a**) $P_l$ = 0.01 mW, (**b**) $P_l$ = 0.1 mW, (**c**) $P_l$ = 1 mW and (**d**) $P_l$ = 10 mW. The other parameters are the same as those in Figure 2.

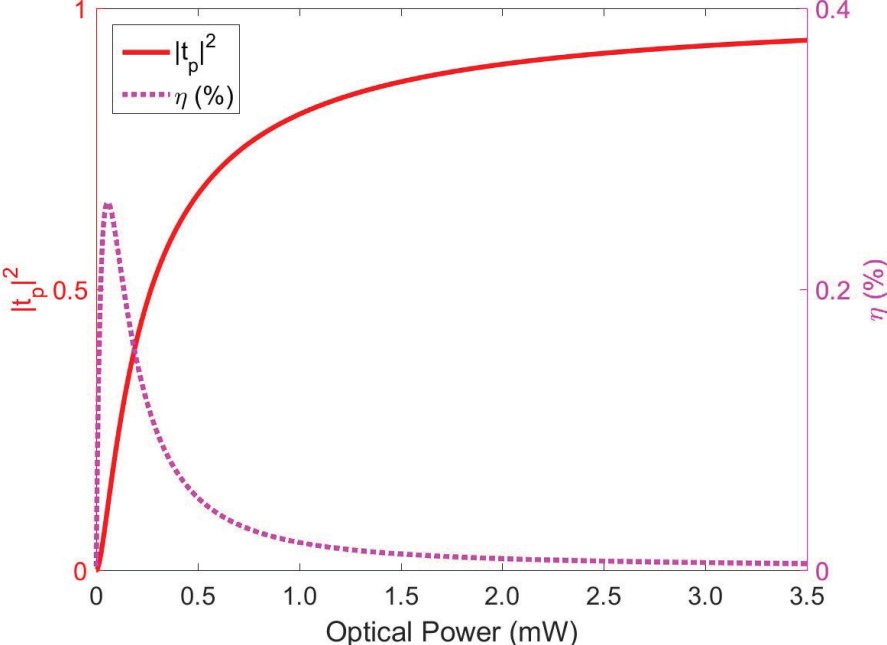

**Figure 4.** The efficiency of the first-order sideband generation $|t_p|^2$ and the third-order sideband generation $\eta(\%)$ vary with the power of the control field under the resonance condition $\Omega = \Omega_m$. The red solid curve represents the first-order sideband $|t_p|^2$ and the pink dotted line represents the third-order sideband $\eta$, respectively. The other parameters are the same as those in Figure 2.

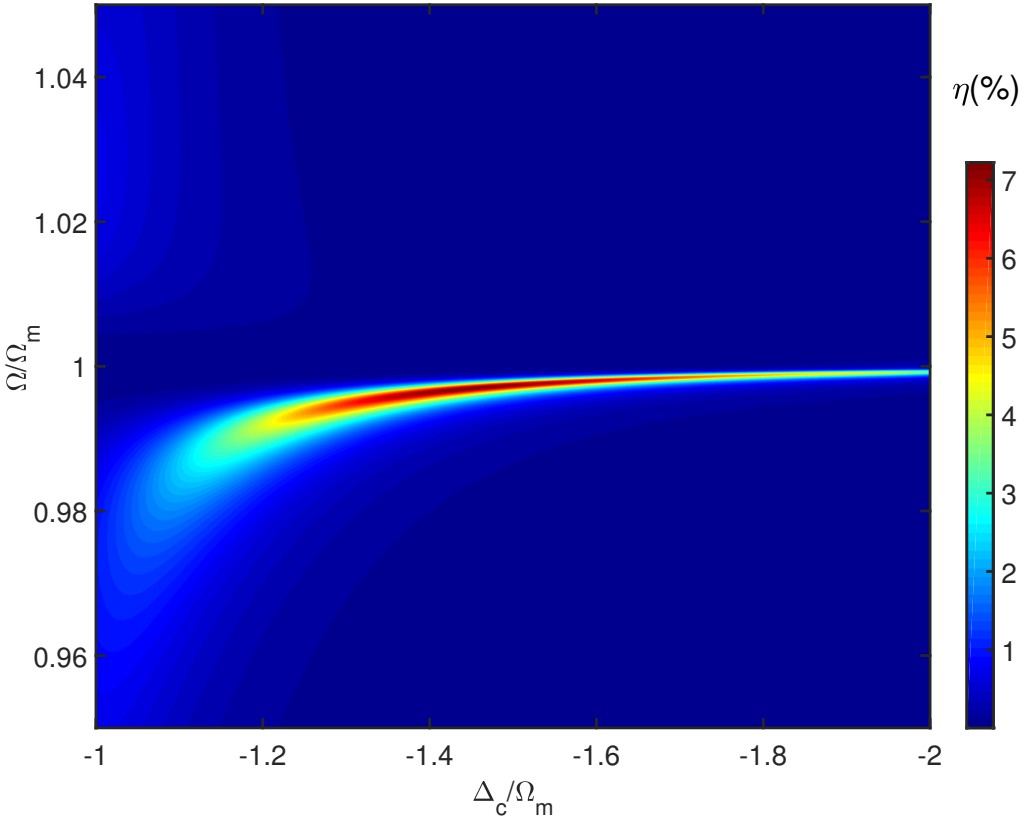

**Figure 5.** The efficiency of the third-order sideband generation $\eta(\%)$ as a function of the detuning $\Delta_c/\Omega_m$ and the beat frequency $\Omega/\Omega_m$. The power of the control field is $P_l = 5$ mW and the other parameters are the same as those in Figure 2.

The effective amplitude of the third-order sideband reaches the maximum, about 7%, under the condition $\Delta_c \approx -1.2\Omega_m$. Further increasing the control field detuning, however, does not lead to a significant improvement; instead, there a slight decline, and the amplitude of the third-order sideband is stable around 6.5% under the resonance condition $\Omega = \Omega_m$. Interestingly, we observe that the relationship between $\eta$ and the driving field frequency $\Omega$ appears to be totally different. More specifically, for the case of $\Omega > \Omega_m$, the effective amplitude of the third-order sideband is quite low and the change of the control field detuning $\Delta_c$ does not improve the effective amplitude of the third-order sideband. However, for the case of $\Omega < \Omega_m$, the effective amplitude of the third-order sideband increases as the control field detuning $\Delta_c$ increases. The physical mechanism can be understood as follows: the asymmetry of sideband distribution is caused by the constructive and destructive interference between the direct third-order sideband process and the upconverted first-order sideband process. Therefore, we can simultaneously adjust the driven frequency $\Omega$ and the control field detuning $\Delta_c$, to achieve the largest amplitude of the third-order sideband rather than by relying on a strong laser drive [46,47].

Due to the asymmetry of the third-order sideband generation around the resonance condition $\Omega = \Omega_m$, it is necessary for us to study the specific driven frequency $\Omega$ that satisfies the third-order sideband increase by adjusting the detuning $\Delta_c$. As shown in Figure 6, the blue dotted line represents the variation of the amplitude of the third-order sideband with the control field detuning $\Delta_c$ under the driven frequency $\Omega/\Omega_m = 0.994$. We find that the efficiency of the third-order sideband $\eta$ with the change of $\Delta_c$ satisfies the Lorentz-like distribution. As the blue dotted line shows, the effective amplitude of the third-order sideband increases with raising the detuning and reaches the maximum value, about 5.8%, under $\Delta_c = -1.27\Omega_m$. However, the efficiency $\eta$ decreases when we continue to enhance the control field detuning $\Delta_c$ and tends to stablilize. Next, we consider another specific driving frequency $\Omega/\Omega_m = 0.996$. The correlation between the process of

the third-order sideband generation $\eta$ and the detuning $\Delta_c$ is the same as $\Omega/\Omega_m = 0.994$ (the red solid line shows), and the maximum value of $\eta$ is about 7% at $\Delta_c = -1.4\Omega_m$. Likewise, the same result as the pink dash-dotted line shows for $\Omega = 0.998\Omega_m$, and the maximal efficiency of the third-order sideband reaches 6.3% at $\Delta_c = -1.62\Omega_m$. The physical mechanism by which third-order sideband generation efficiency has a local minimum at resonance ($\Omega = -\Delta_c = \Omega_m$) can be understood as follows: the upconverted first-order sideband process is weak when the OMIT occurs and when the detuning $\Delta_c$ of the driving field was increased, one of the absorption peaks will consequently move. At that time, the efficiency of $|t_p|^2$ will quickly reduce near the resonance condition $\Omega = \Omega_c$; thus, it results in a visible third-order sideband generation. From the above discussion, one can achieve the maximum values of the third-order sideband by regulating driven frequency $\Omega$ and detuning $\Delta_c$ in the practical application of precision measurement.

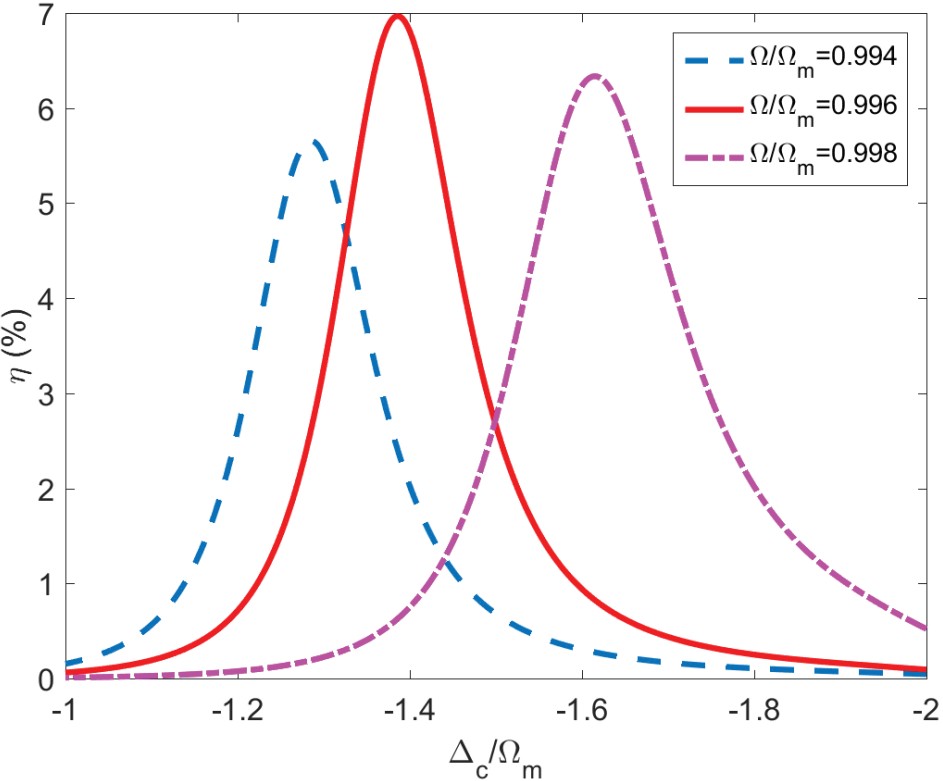

**Figure 6.** The efficiency of the third-order sideband generation $\eta\,(\%)$ varies with the detuning $\Delta_c/\Omega_m$ under different beat frequency $\Omega$. The blue dashed line, red solid line, and pink dotted line indicate the beat frequency $\Omega/\Omega_m$ = 0.994, 0.996, and 0.998, respectively. The power of the control field is $P_l$ = 5 mW and the other parameters are the same as those in Figure 2.

## 4. Conclusions

In summary, we investigate the third-order optomechanical nonlinearity by using a perturbative approach, and give an analytical solution of third-order sideband generation. In addition, an effective method to enhance the high-order sideband generation by adjusting the detuning of a laser field is proposed, rather than relying on a strong laser drive. The advent of research into high-order optomechanical nonlinearity may have important implications for studying the behaviors of the mechanical effect of light, and furthermore, is anticipated to bring in a wealth of applications, especially offering a more accurated nonlinear optical method for precision measurements.

**Author Contributions:** Q.W.: Carried out the calculations, Wrote the main manuscript text, Prepared all figures, Reviewed the manuscript, Contributed to the interpretation of the work, Writing of the manuscript. H.-J.S.: Carried out the calculations, Participated in the discussions, Reviewed the manuscript, Contributed to the interpretation of the work, Wrote of the manuscript. All authors have read and agreed to the published version of the manuscript.

**Funding:** This work was supported by the Doctoral Program of Guangdong Natural Science Foundation, China (Grant No. 2018A030310109), the Doctoral Project of Guangdong Medical University (Grant No. B2017019).

**Institutional Review Board Statement:** Not applicable.

**Informed Consent Statement:** Informed consent was obtained from all subjects involved in the study.

**Data Availability Statement:** The data that support the findings of this study are available from the corresponding author upon reasonable request.

**Conflicts of Interest:** The authors declare no conflict of interest.

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
