# Peer review of "Higher-Order Optomechanical Nonlinearity Based on the Mechanical Effect of Light"

_photonics, doi:10.3390/photonics10090991_

Round 1

Reviewer 1 Report

This paper is well-presented in the theoretical model and simulation results, which I would suggest to accept this paper in its current form.

Minor editing English language.

Author Response

Responses to the comments of Referee #1

Comment #. This paper is well-presented in the theoretical model and simulation results, which I would suggest to accept this paper in its current form.

Reply #. We thank the Referee for these positive comments and we are glad to know that he/she thinks that our manuscript is suitable for publication in photonics.

Reviewer 2 Report

This is an interesting and original article but it can be improved in order to be accepted for publication.

1. It is usual that the last paragraph of the introduction should contain a resume of each one of the following sections.  This is missing.

2.- All the figure captions must be improved and extended. They are too short and important information needed by a potential reader is missing. In particular all symbols should be included and coincide with what each picture is showing.

3.-In line 149/150 (also in 176) you must justify and explain the values used in your simulations.

4.- In 176 you make reference to experimental parameters but there are not experiments carried out but simulations.

5.- In line 181 it is mentioned that this is "as expected".  It would be better to carefully explain why of this.

6.- In line 184 a not deep opacity window is mentioned but it is not explained why of this.

7.- Along the article there is an abuse of the word "obvious" which should be avoided.

8.- in line 250 it is mentioned that "...an effective method to enhance the high-order side band generation by adjusting the detuning of laser field rather than a strong laser drive is proposed".  It would be good to include any experimental evidence you may have to support this statement.

no comments

Author Response

Responses to the comments of Referee #2

Comment # 1. This is an interesting and original article but it can be improved in order to be accepted for publication.

Reply # 1. We thank the Referee for acknowledging the quality of our work, finding our results and the manuscript suitable for publication in photonics, and for valuable suggestions that have helped us improve the manuscript. We have modified our manuscript according to his/her suggestions. Below, we provide a point-by-point response to the referee’s comments.

Comment # 2. It is usual that the last paragraph of the introduction should contain a resume of each one of the following sections. This is missing.

Reply # 2. We thank the Referee for this useful suggestion. In the revised manuscript, we have added a resume about the various parts of the article. Please see lines 88-96 in red in the revised manuscript.

Comment # 3.- All the figure captions must be improved and extended. They are too short and important information needed by a potential reader is missing. In particular all symbols should be included and coincide with what each picture is showing.

Reply # 3. We thank the Referee for this helpful suggestion. In the revised manuscript, we have improved and expanded all the picture captions and explained all the symbols that appear in the pictures. Please refer to the red contents in the picture caption of the revised manuscript.

Comment # 4.-In line 149/150 (also in 176) you must justify and explain the values used in your simulations.

Reply # 4. We thank the Referee for this helpful suggestion. In the revised manuscript, we have explained the parameter values used in the simulation one by one. Please see lines 182-188 in red in the revised manuscript.

Comment # 5.- In 176 you make reference to experimental parameters but there are not experiments carried out but simulations.

Reply # 5. We thank the Referee for this suggestion. In the revised manuscript, we have added references to experimental articles. Please see lines 211 in the revised manuscript.

Comment # 6.- In line 181 it is mentioned that this is "as expected". It would be better to carefully explain why of this.

Reply # 6. We thank the Referee for this helpful comment. In the revised manuscript, we have explained why increasing the power of the driving field leads to an increase in the efficiency of third-order sideband generation. For more detail, please see lines 218-219 in red in the revised manuscript.

Comment # 7.- In line 184 a not deep opacity window is mentioned but it is not explained why of this.

Reply # 7. We thank the Referee for this helpful comment. In the revised manuscript, we have explained the appearance of the opacity windows. For more detail, please see lines 221-228 in red in the revised manuscript.

Comment # 8.- Along the article there is an abuse of the word "obvious" which should be avoided.

Reply # 8. We thank the Referee for this kind suggestion. In the revised manuscript, we use the word "obvious" carefully to avoid its abuse.

Comment # 9.- in line 250 it is mentioned that "...an effective method to enhance the high-order sideband generation by adjusting the detuning of laser field rather than a strong laser drive is proposed". It would be good to include any experimental evidence you may have to support this statement.

Reply # 9. We thank the Referee for this useful comment. In the revised manuscript, we cite relevant experimental articles to support our statement, i.e., references 47 and 48 in the manuscript. [Phys. Rev. Lett. 127, 134301 (2021), Phys. Rev. Lett. 128, 153901 (2022).]

Reviewer 3 Report

A very nice piece of theoretical research - well suited to the journal!

Just some minor edits are needed:

1) Eq on page 5 above Eq 9 needs to be broken over 2 lines to avoid going into the margin

2) Figure 2 and 4 color bars needs to be labelled

3) How would results change in a master equation approach was used?

Mostly fine

Author Response

Responses to the comments of Referee #3

Comment # 1. A very nice piece of theoretical research - well suited to the journal! Just some minor edits are needed:

Reply # 1. We thank the Referee for these positive comments and we are glad to know that he/she thinks that our manuscript is suitable for publication in photonics. In the revised manuscript, we have modified our manuscript according to his/her suggestions. Below, we provide a point-by-point response to the referee’s comments.

Comment # 2. Eq on page 5 above Eq 9 needs to be broken over 2 lines to avoid going into the margin.

Reply # 2. We thank the Referee for this kind suggestion. In the revised manuscript, we have adjusted the layout of the Eq 9 to avoid entering the margin.

Comment # 3. Figure 2 and 4 color bars needs to be labelled.

Reply # 3. We thank the Referee for this useful comment. In the revised manuscript, we have labelled the color bars in Figure 2 and Figure 4.

Comment # 4. How would results change in a master equation approach was used?

Reply # 4. We thank the Referee for this professional comment. Indeed, in weakly driven cavity optomechanical system, many interesting quantum phenomena can be obtained by solving the master equation of the system, such as traditional and non-traditional photon blocking effect [Phys. Rev. Lett. 104, 183601 (2010), Phys. Rev. A 87, 013839 (2013)]. In our work, we consider a strongly driven cavity optomechanical system and the dynamics corresponding to the Hamiltonian and damping of the oscillator modes into independent thermal baths can be described by a system of linear Heisenberg-Langevin equations in a semiclassical approximation. Therefore, we did not use the master equation approach in our work. We believe that this will be an interesting and important topic that we will pay attention to in our future work.

Round 2

Reviewer 2 Report

The revised manuscript takes into account the comments and observations stated in the previous revision.  I recommend to Accept in Present form.